# Development of the Rehabilitation Health Policy, Systems, and Services Research Field: Quantitative Analyses of Publications over Time (1990–2017) and across Country Type

**DOI:** 10.3390/ijerph17030965

**Published:** 2020-02-04

**Authors:** Tiago S. Jesus, Helen Hoenig, Michel D. Landry

**Affiliations:** 1Global Health and Tropical Medicine & WHO Collaborating Center on Health Workforce Policy and Planning, Institute of Hygiene and Tropical Medicine-NOVA University of Lisbon, 1349-008 Lisbon, Portugal; 2Physical Medicine and Rehabilitation Service, Durham Veterans Administration Medical Center, Durham, NC 27705, USA; Helen.Hoenig@va.gov; 3Division of Geriatrics, Department of Medicine, Duke University Medical Center, Durham, NC 27710, USA; 4School of Medicine, Duke University, Durham, NC 27710, USA; mike.landry@duke.edu; 5Duke Global Health Institute, Duke University, Durham, NC 27710, USA

**Keywords:** health policy, health services research, rehabilitation, PubMed, publications

## Abstract

Background: Health policy, systems and services research (HPSSR) is increasingly needed to enable better access to, and value of, rehabilitation services worldwide. We aim to quantify the growth of Rehabilitation HPSSR publications since 1990, compared to that of overall rehabilitation research and overall HPSSR. Methods: Quantitative, comparative analysis of publication trends using the PubMed database and its indexation system. Comprehensive search filters, based on Medical Subject Headings (MeSH), were built and calibrated to locate research articles with content on HPSSR and rehabilitation of physical impairments. Additional filters were used for locating research publications declaring funding support, publications in rehabilitation journals, and finally publications focused on high-income (HICs) or low- and middle-income countries (LMICs). The same approach was used for retrieving data on comparator fields—overall HPSSR and overall rehabilitation research. Linear regressions, with ANOVA, were used for analyzing yearly publication growths over the 28-year time frame. Results: Rehabilitation HPSSR publications in PubMed have grown significantly from 1990 to 2017 in the percentage of all rehabilitation research (from 11% to 18%) and all HPSSR (from 2.8% to 3.9%; both *p* < 0.001). The rate of Rehabilitation HPSSR published in rehabilitation journals did not change significantly over time (*p* = 0.47). The rates of publications with declared funding support increased significantly, but such growth did not differ significantly from that of the comparator fields. Finally, LMICs accounted for 9.3% of the country-focused rehabilitation HPSSR since 1990, but this percentage value increased significantly (*p* < 0.001) from 6% in 1990 to 13% in 2017. Conclusion: Rehabilitation HPSSR publications, i.e., those indexed in PubMed with related MeSH terms, have grown in both absolute and relative values. Rehabilitation HPSSR publications focused on LMICs also grew significantly since 1990, but still remained a tiny portion of the Rehabilitation HPSSR publications with country-specific MeSH terms.

## 1. Introduction

Rehabilitation services are increasingly needed around the globe, following the global ageing population and the increased rates of chronic conditions and disabilities [1,2]. However, health systems worldwide are often under-resourced and ill-prepared to efficiently meet the rehabilitation needs of the population [3,4,5,6,7,8,9]. Production, dissemination, and uptake of rehabilitation-focused health policy, systems, and services research (i.e., Rehabilitation HPSSR) are a means to strengthen health systems’ capacity to deliver appropriate, high-value rehabilitation services, toward sustainably meeting the growing rehabilitation needs of the population [10,11,12,13,14].

Differing from clinical or basic research, health policy and systems research addresses the upstream aspects of health, organizations, and policies [15,16,17]. It does it so as an inter-disciplinary field that seeks to understand and find solutions as to how: (1) societies and stakeholders organize themselves to achieve collective health goals; (2) health systems respond and adapt to health policies, and (3) health policies can shape and are shaped by health systems and broader determinants of health [15,17]. Health policy and systems research address any, and often multiple, ‘building blocks’ of health systems (e.g., governance, workforce, financing), within a systems-thinking approach toward promoting the coverage, quality, efficiency and equity of health systems [15,18,19,20,21]. At the service-delivery level, and with increasing convergence with a broader health systems research [16,19,22], health services research refers to a multidisciplinary field of investigation on how social factors, financing, organizations, technologies, innovation, and personal behaviors affect the access to healthcare, its quality, and its cost [23,24]. Taken together, HPSSR addresses aspects within or at the interface of health policy, of health system’s organization, and of healthcare delivery - for improved healthcare access, healthcare value, and population health.

In the rehabilitation field, advancing HPSSR is an increasing priority. This has been evident through recent publications in scientific journals on the role for and value of advancing rehabilitation health-services research [10,11,25,26,27], a number of empirical works, research syntheses and frameworks on rehabilitation health policy and health systems issues [12,28,29,30], rehabilitation stakeholders or interest groups with a focus on advancing rehabilitation services and services research [31,32], funding mechanisms or funding policies welcoming HPSSR in the rehabilitation field [33,34], and the recent input from the World Health Organization for developing a rehabilitation health policy and health systems research agenda [13].

Despite this growing interest, to our knowledge there is no systematic, quantitative analysis of the evolving research production on Rehabilitation HPSSR topics, even though its exists for other rehabilitation research areas [35,36,37,38,39] and for aspects of the overall HPSSR [21,40,41,42,43,44,45]. 

Hence, we aim to quantify the growth in the number of Rehabilitation HPSSR publications from 1990 and 2017, compared to that of overall rehabilitation research and overall HPSSR. This is operationalized through the following study questions:What is the total amount, absolute growth, and relative growth of Rehabilitation HPSSR publications (i.e., as the percentage of all rehabilitation research publications, and as the percentage of all HPSSR publications)?Which portion of the Rehabilitation HPSSR has been published in rehabilitation journals, and has that changed over time?Which portion of the country-specific rehabilitation HPSSR publications focus on high-income countries (HICs) versus low- and middle-income countries (LMICs)?What are the yearly rates of the Rehabilitation HPSSR publications that declare funding support, and how does that compare to rehabilitation research overall and HPSSR overall.

Answering these questions will provide information on crude trends in HPSSR articles in the rehabilitation literature from 1990 to 2017.

## 2. Materials and Methods

### 2.1. Design

Quantitative, comparative analysis of publication trends over time using data from the indexation system of a large health research database. 

### 2.2. Data Source

The PubMed database (http://www.ncbi.nlm.nih.gov/pubmed) and the indexation system for its largest sub-component (i.e., MEDLINE) was used for this analysis. Although not exhaustive, PubMed is a comprehensive research database. For instance, there is evidence that PubMed contains about 85% of the studies included in health-based systematic reviews [46,47]. Similarly, in a sample of randomized controlled trials relevant to physical therapy, PubMed indexed 89% of them, slightly more than EMBASE (88%) and close to PEDro (92%) and CENTRAL (95%). Of note, PEDro and CENTRAL are databases specialized in physical therapy and clinical trials, respectively. PubMed has other advantages for this study—it is freely-accessible, hence the study is easily reproducible, and it addresses the health field as a whole, enabling comparative analyses. 

PubMed/MEDLINE is suited for analyzing publication trends due its comprehensive indexation system, based on Medical Subject Headings (MeSH) that are organized within a hierarchical tree and assigned to each paper by trained indexers. Articles are therefore systematically indexed by research topic and methodology regardless specific words authors have used. Using such capabilities, PubMed has been a source for large analyses of publication trends within [35,36,37,39,48] and outside [49,50,51] the rehabilitation field. Here, we also relied fully on the PubMed indexing facilities (i.e., do not manually screen or review titles, abstracts or full texts) to identify rehabilitation HPSSR publications and those of comparator fields: overall HPSSR and overall rehabilitation research. 

### 2.3. Search Filters–Construction

We used a mixed-methods approach for a comprehensive determination of search filters, i.e., combinations of relevant MeSH terms, for each component of our searches (e.g., HPSSR, rehabilitation, journal and country types).

#### 2.3.1. HPSSR

For building the HPSSR filter, we first relied on working definitions and conceptual papers within the HPSSR arena [13,15,16,17,18,19,20,23,24]. Then, the key concepts extracted were translated into MeSH terms, using target and snowballing searches in the MeSH database [52]. For example, like other research [21], we searched the MeSH database for key terms around the six ‘building blocks’ of health systems (i.e., governance/leadership, workforce, financing, information systems, etc.). Additionally, we searched the MeSH database for other related key concepts (e.g., health planning, services supply and demand, health equity, quality improvement, healthcare access, etc.). The non-redundant terms (i.e., all of those not fully contained within upper-level MeSH terms indeed included) were combined as alternatives to one another (i.e., using the Bolean operator “OR”). We used the tag [majr] for each term, to retrieve only articles with any of those MeSH terms indexed as a Major Topic. This means that we selected only the research focused on a HPSSR topic, not those merely related to it. 

#### 2.3.2. Rehabilitation

We focused on the rehabilitation of people with physical impairments. Hence, some forms of rehabilitation were excluded, such as the rehabilitation of mental health conditions (psychiatric rehabilitation), substance abuse, people with sensory impairments per se (e.g., correction of hearing impairment), and people with oral health conditions. However, for example, the rehabilitation of cognitive, communicative, and neuro-behavioral impairments as a result of or associated to physical impairments were included in the scope of rehabilitation covered [1,37,53]. With that field restriction, and based on existing conceptual works or definitions of rehabilitation [1,53,54,55,56,57] as well as published search filters [35,36,37], we also approached the MeSH database to identify rehabilitation-related MeSH terms and any to be excluded with the Bolean operator “NOT”; for example, “psychiatric rehabilitation” [MeSH] is within the upper-level “rehabilitation”[MeSH], hence needed to be actively excluded. As above, we only included non-redundant MeSH terms, selected as a Major Topic. The exclusions were made with regular MeSH terms. 

#### 2.3.3. Research Publications in Humans

PubMed contains many publication types (e.g., editorial, letters, commentaries) that would not be eligible as research publications. Hence, based on previous search filters [35,36,37] and target searches in the MeSH database, we constructed a search filter for locating all kinds of empirical-based research publications, including qualitative research, case reports, systematic reviews, or practice guidelines. As systematic reviews have only been indexed as such in PubMed since 2019, we used a combined set of search terms to locate them, including non-MeSH terms [36,37] Finally, we narrowed down the research publications for those on “humans” subjects (e.g., not on animals). This filter applied to all the searches, including those targeting comparator fields. 

#### 2.3.4. Journal Type

We used the National Library of Medicine’s journal catalog to identify journals that were indexed in PubMed and fit the scope of “physical medicine and rehabilitation” [sb], which includes, for example, physical and occupational therapy. As a result of the search, 116 journals were identified and the respective MeSH terms were retrieved as alternative to one another.

#### 2.3.5. Funding Support

We use “Support of Research” [Publication Type] as the overarching MeSH term for articles declaring funding support from any venue. 

#### 2.3.6. Country-Type

We used the World Bank’s classification to determine which countries and territories (e.g., Hong Kong) are high- or low-income, then translated these to the respective MeSH terms, if existent (i.e., not for a minority of small territories). These terms were set as alternative to one another (using the operator “OR”) for the groups of HICs and LMICs. Of note, not all articles addressed specific countries (e.g., many systematic reviews), and articles could address no particular countries or HICs and LMICs at the same time. Also, when specific countries were addressed but this was not be explicit in the articles’ title or abstract, often the articles were not granted a country-specific MeSH term. Hence, the amount of country-specific rehabilitation HPSSR is not necessarily equal to that of overall rehabilitation HPSSR.

### 2.4. Search Filters–Calibration

For calibrating the search filters, we ran pilot searches with or without some of the MeSH terms the research team appraised as at the borderline for inclusion or exclusion, assessing the impact they had on the search results. For example, we tested the collective impact of introducing a set of MeSH terms focused on assistive devices (“Self-Help Devices”[Majr] OR “Exoskeleton Device”[Majr] OR “Artificial Limbs”[Majr] OR “Orthotic Devices”[Majr] OR “Canes”[Majr] OR “Walkers”[Majr] OR “Crutches”[Majr]) within the search filter for rehabilitation. Including this whole set of terms added less than 2% more Rehabilitation HPSSR articles retrieved. Hence, the inclusion of this set was not deemed to carry substantial risk to preciseness (e.g., inclusion of too many articles not directly related to the rehabilitation: issues on device manufacturing, etc.), with likely benefits for increased sensitivity (e.g., identifying additional articles on the training or prescription of assistive devices). Theoretically, HPSSR cares about making medical products (including assistive technologies) available to those who need them [58]. Therefore, we retained this set of search terms in our final search strategy.

A final calibration exercise entailed a random selection of 30 articles from the authors’ personal resources, published before 2017, that were deemed relevant for the field of Rehabilitation HPSSR by the research team. These articles were searched for in PubMed, and their major MeSH term retrieved. This whole process enabled the identification of one more relevant MeSH term for the rehabilitation search filter (i.e., “recovery of function” [MeSH]) and one more relevant term for the filter on research publications. For instance, some research articles (e.g., using mixed methods) were not indexed for any method/publication type in particular, but were indexed for “research support” [publication type]. Hence, we added this term to the research publications filter, assuming that all articles that declare “research support” are indeed research. This resulted in a substantial increase in the number of detected publications, sometimes over one-third more, with the added sensitivity coming with low risk to preciseness. 

Appendix A provides the full set of calibrated search filters that we finally used, and the details of how these were combined in PubMed searches to answering our study questions. 

### 2.5. Searches Conduct and Time Span

The searches were conducted in late October 2019 but were narrowed down in publication dates from January 1, 1990 to December 31, 2017. This accounted for the fact that “publication types” have been systematically indexed in PubMed only since 1990 and for the typical 2-year delay in the PubMed indexation. 

Specifically, for the searches focused on funding support, the time span was narrowed down to end at December 2015. Indeed, in preliminary data visualizations, we observed outlier results (substantially lower number of publications for the years 2016 and 2017) in every search, across the fields, i.e., focused or not on rehabilitation or HPSSR. This suggests a delay or relevant change in this indexation process in particular, rather than reflecting changes in publications data itself. Hence, data on 2016 and 2017 for declared funding support were excluded.

### 2.6. Data Extraction, Management and Analysis

From each search in PubMed, the total and yearly volume of publications were exported to Excel (Microsoft Corporation). There, we computed relevant percentages (e.g., percentage of Rehabilitation HPSSR publications among all rehabilitation research publications, overall and for each year). As this paper involved comparative analyses, the data were often analyzed and reported in those relative percentages. Run charts and then linear, exponential or logarithmic regression models, according to best fit (determined by *r*^2^ values and visual examination) were used to analyze any changes in the count data and computed percentages over time. We retained the linear models whenever the fit was similar (e.g., *r*^2^ values difference <0.02).

We used simple linear regressions, with ANOVA, for determining whether any yearly changes, in counts or percent values, were significant. We use the JASP 0.10.20 software (University of Amsterdam, Amsterdam, The Netherlands) for this analysis. *p* values <0.05 were considered statistically significant.

## 3. Results

Below we present the results according to our initial research questions.

### 3.1. Absolute and Relative Growth

From 1990 to 2017, a total of 23,105 Rehabilitation HPSSR publications were found, with a significant linear growth of 63.3 publications a year (95% Confidence Interval (CI): 58.2–68.4; *p* < 0.001; *r*^2^ = 0.96).

For the relative growth, Figure 1 shows that HPSSR publications accounted for 17.6% of the rehabilitation research publications in 2017, compared to 10.6% back in 1990. This equates to a significant yearly growth (*p* < 0.001), albeit logarithmic in type (*r*^2^ (log model) = 0.83), i.e., more pronounced in the early years.

Figure 2, in turn, shows that rehabilitation accounted for 3.9% of all HPSSR publications in 2017, compared to only 2.8% in 1990, referring to a significant and linear growth in this relative value (*p* < 0.001; *r*^2^ (linear model) = 0.9).

### 3.2. Journal Type

Figure 3 shows that the percentage of Rehabilitation HPSSR published in rehabilitation journals did not significantly change over time (*p* = 0.49), erratically moving around 24%. On the other hand, the percentage of the overall rehabilitation research (i.e., not only the Rehabilitation HPSSR) that was published in rehabilitation journals increased significant yet logarithmically over time (*p* < 0.001; *r*^2^ (log model) = 0.55), from 17% in 1990 to approaching 22% in 2017, the typical values of the Rehabilitation HPSSR. 

### 3.3. HICs Versus LMICs

From 1990 to 2017, a total of 9733 Rehabilitation HPSSR publications were indexed in PubMed with a focus on any HIC, and 986 on any LMIC. This means that LMICs account for up to 9.3% of the Rehabilitation HPSSR publications that had country-specific MeSH terms; however, Figure 4 shows that such a percentage value has been growing significantly over time (*p* < 0.001), from 6% in 1990 to 13% in 2017.

### 3.4. Funding Support Declared

From 1990 to 2015, 11,037 Rehabilitation HPSSR publications declared funding support, i.e., 47.8% of them. Figure 5 shows that the proportion of publications declaring support increased significantly over time, linearly from 43% (1990) to 58% (2015) (*p* < 0.001; 95% CI: 58.2–68.4; *r*^2^ = 0.96). This growth was comparable to (i.e., 95% CIs partly overlapped) rehabilitation research as a whole and HPSSR as a whole, with all three approaching 2015 with values within the 53–58% range.

## 4. Discussion

Rehabilitation HPSSR publications, notably those indexed for rehabilitation and HPSSR topics in the PubMed database, have growth significantly from 1910 to 2017, in both absolute and relative values. Indeed, the growth of the rehabilitation HPSSR publications significantly outpaced that of both comparators, i.e., grew significantly in the percentage of all rehabilitation research, and in the percentage of all HPSSR. We also observed discrepancies in the Rehabilitation HPSSR publications focused on HICs versus LMICs, with the latter accounting for 9.3% of the Rehabilitation HPSSR publications that had at least a country-specific MeSH term. However, a significant, growing trend from 6% to 13% (1990–2017) was observed in that percentage value. Below we present five points of discussion related to our findings.

First, the observed growth of publications on HPSSR topics within the overall rehabilitation research, i.e., from 11% to 18% (1990–2017), means that rehabilitation stakeholders are conducting, comparatively, more HPSSR. Nonetheless, HPSSR topics still represent less than one-fifth of all rehabilitation research publications and the observed relative growth was less pronounced (i.e., flattening) in more recent years. It is hard to know what is the desirable figure with regard to the proportion of rehabilitation research that should be devoted to HPSSR; however, what we do know is that Rehabilitation HPSSR helps to enhance the value or population impact of other forms of rehabilitation research (e.g., clinical rehabilitation research). For example, HPSSR would help to translate the growing clinical knowledge into more common, more accessible, and more affordable rehabilitation practices, boost innovation and accountability for service organization and delivery, and make generalizable knowledge responsive or adaptive to identified local needs [10,11,15,59,60].

Second, Rehabilitation HPSSR represents a significantly increased proportion of publications within HPSSR, from 2.8% in 1990 to 3.9% in 2017. The growth over time has been significant and relatively linear, i.e., constant, over the 28-year period; nonetheless, rehabilitation represents a very small portion of all HPSSR. This may be concerning given the current high levels of physical rehabilitation needs worldwide (e.g., in 2017, 40% of the world’s non-fatal health loss came from conditions amenable to physical rehabilitation [1]), the significant growth in those needs observed across country and condition types [1,61], the evidence for large gaps in the physical rehabilitation supply [6,29], and the growing evidence on the individual, economic, and broader societal benefits associated to the delivery of appropriate physical rehabilitation services [2,62,63,64,65]. In other words, we have found that the amount of Rehabilitation HPSSR is a disproportionate to the global population’s needs for such services. All accounted for, we suggest that there is a urgent need to address the collective negligence and limited development of rehabilitation resources within and across many countries [2,6,8,9], the gaps in quality, access, and value of rehabilitation care worldwide [5,6,8,9,14], and the growing unmet needs for rehabilitation across countries of all income levels [1,6]. 

Third, LMICs accounted for a relatively marginal share of the country-specific Rehabilitation HPSSR publications, i.e., less than 10% for the all 28 years analyzed. This diverges, for example, from the overall health policy and systems research, which is close to a 50–50% distribution between HICs and LMICs [21], with publications relevant to LMICs being those increasing at the highest rates [40]. The seminal World Report on Disability [66] estimated that 80% of people with disabilities live in LMICs, a recent analyses of data from the Global Burden of Disease 2017 found that LMICs account for 77% of the physical rehabilitation needs [1], while several reports point towards large unmet rehabilitation needs in LMICs [5,6,8,9]. While we found a significant growing trend in the portion of the country-specific Rehabilitation HPSSR publications that were focused on LMICs(up to 13% in 2017),a still higher preponderance of rehabilitation HPSSR is likely needed, especially in LMICs; transformational rather than incremental improvements would be required to address this gap. Capacity would need to be created, for example, in developing a research workforce with the means, focus, and capabilities to implement a Rehabilitation HPSSR agenda worldwide [10,22,67,68,69], including with a special focus on LMICs [70,71,72]. 

Fourth, we did not observe a significant change in the rate of Rehabilitation HPSSR publications in rehabilitation journals. Given that we found that less than 25% of rehabilitation HPSSR publications are in rehabilitation journals and given that we lack databases or repositories specific to Rehabilitation HPSSR, persons wanting to locate and use Rehabilitation HPSSR evidence will need to rely on comprehensive search strategies in large health databases. 

Finally, we found a significant increase in the rates of Rehabilitation HPSSR declaring funding support and those rates are comparable to similar research. This may be comforting for the rehabilitation research community; however, this finding occurs against the backdrop of overall stagnation or reduction in funding for health services research in the US [45], with US health services research representing a mere 0.3% of the total US healthcare expenditures or only 5% of all research funding [73]. While in low-income countries the funding availability for the overall health policy and systems research has increased, notably to institutions in Sub-Saharan Africa [21], we found the number of publications focused on LMICs was insufficient to enable an analysis of the funding trends over time on the Rehabilitation HPSSR focusing on LMICs.

### Limitations

This paper has several limitations:

We included only research publications indexed in PubMed, not the full spectrum of research publications. Similarly, we fully relied on the PubMed indexation system for this large dataset analysis, while the indexation system is not fully sensitive and precise. Nonetheless, the PubMed indexation refers to a standard process conducted by trained indexers. Also, the same search terms, comprehensively developed, were applied across the comparator fields. As the results are essentially reported in relative percentages, this makes the comparative results less prone to bias. Furthermore, due to typical delays in the PubMed indexation, we could only present results up to 2017 (and those for funding only to 2015); hence, the effects of more recent pushes for advancing the Rehabilitation HPSSR [10,11,13,34] may not be noticed yet. It is also worth noting that we analyzed merely the number of research publications, not their methodological type or quality. Besides, we were unable to analyze the yearly evolution of the Rehabilitation HPSSR in LMICs that have had declared funding support, given the low number of publications per year. Furthermore, we did not analyze the country of origin of the authors or the authors’ network [42]. Additionally, only about half of the Rehabilitation HPSSR publications in PubMed had a country-specific MeSH term (e.g., were systematic reviews or did not explicitly address a country in the article’s title or abstract); hence the analysis by country-type was carried out over a sample of the Rehabilitation HPSSR publications in PubMed, although this does not preclude a biased sample, i.e., not necessarily favoring HICs or LMICs.

Moreover, we did not analyze the specific scope of Rehabilitation HPSSR by thematic areas [21,44], but solely the whole of these research publications. Similarly, we did not analyze publications from the field of health policy and systems research separately from those in the field of health services research, as both broadly aim to address issues of healthcare access, quality, and value at the population level [16,19,22], thereby it would be complex to select MeSH terms clearly relevant for one but clearly not relevant for the other. Furthermore, we analyzed rates of funding support that research publications declared, not funding support amounts [45]. 

Finally, we focused only on research publications, while perspectives, commentaries, conceptual, narrative or other types of article, e.g., analyzing current situations [74] or describing development or consultation activities [75], have an important role as well. 

Highlighting these limitations are critical in pinpointing the weaknesses of our methodology; however, given the limited availability of the publicly available information, we interpret this study as a starting point, and would invite the rehabilitation and/or HPSSR community to contribute to this discussion.

## 5. Conclusions

This research uses the PubMed indexation capabilities to provide information on crude trends (i.e., publications data not manually screened) on the evolution of the number of Rehabilitation HPSSR articles published from 1990 to 2017. With such a strategy, we have found that Rehabilitation HPSSR publications, i.e., those published and indexed in PubMed with rehabilitation- and HPSSR-related MeSH terms, have grown in both absolute and relative values. However, in 2017, Rehabilitation HPSSR still accounted for merely 18% of the rehabilitation research publications and 4% of the overall HPSSR publications in PubMed. We also found that the portion of Rehabilitation HPSSR publications focused on LMICs, among those that had country-specific MeSH terms, has been growing significantly since 1990, but is still minute, especially given the sizeable unmet needs for rehabilitation services in many LMICs.

## Figures and Tables

**Figure 1 ijerph-17-00965-f001:**
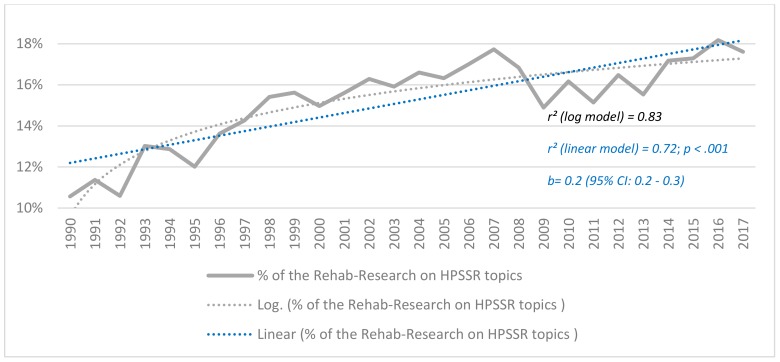
The yearly percentage of the Rehab (i.e., rehabilitation) research that is focused on Health Policy, Systems and Service Research (HPSSR) topics from 1990 to 2017, and the regression model (log: logarithmic) that had the best fit with the data. The *p* value and 95% Confidence Interval (CI) refer to the linear regression model.

**Figure 2 ijerph-17-00965-f002:**
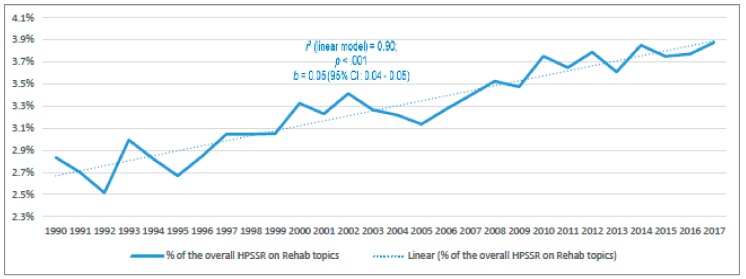
The yearly percentage of the Health Policy, Systems and Service Research (HPSSR) that is focused on Rehab (i.e., rehabilitation) topics from 1990 to 2017, and the regression model (linear) that had the best fit with the data. The *p* value and 95% Confidence Interval (CI) refer to the linear regression model.

**Figure 3 ijerph-17-00965-f003:**
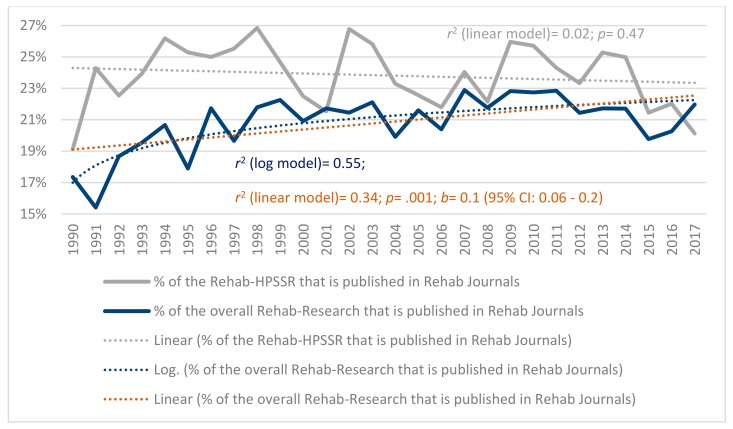
The yearly percentage of the rehab (i.e., rehabilitation) Health Policy, Systems and Service Research (HPSSR) and of the overall rehab research that were published in rehab journals from 1990 to 2017, and the regression models that had the best fit with the data. The *p* values and 95% Confidence Interval (CI) refer to the linear regression models.

**Figure 4 ijerph-17-00965-f004:**
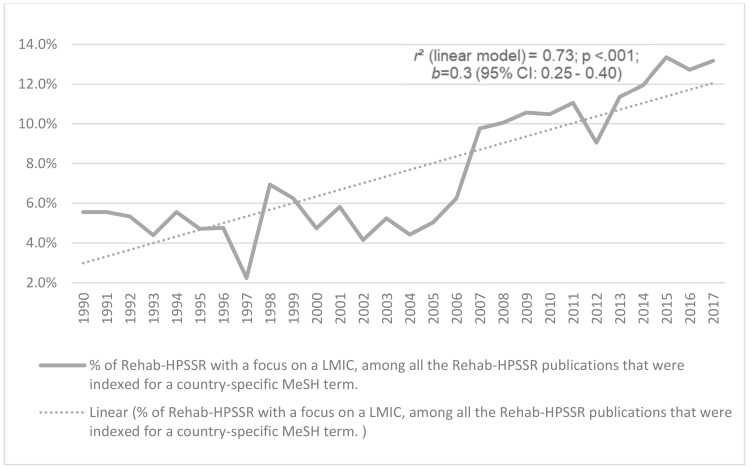
The yearly percentage of the rehab (i.e., rehabilitation) Health Policy, Systems and Service Research (HPSSR) focused on Low- and Middle-Income Income Countries (LMICs) among all rehab-HPSSR publications with a country-specific Medical Subject Heading [1990–2017], and the regression model (i.e. the linear) that had the best fit with the data. The *p* value and 95% Confidence Interval (CI) refer to the linear regression model.

**Figure 5 ijerph-17-00965-f005:**
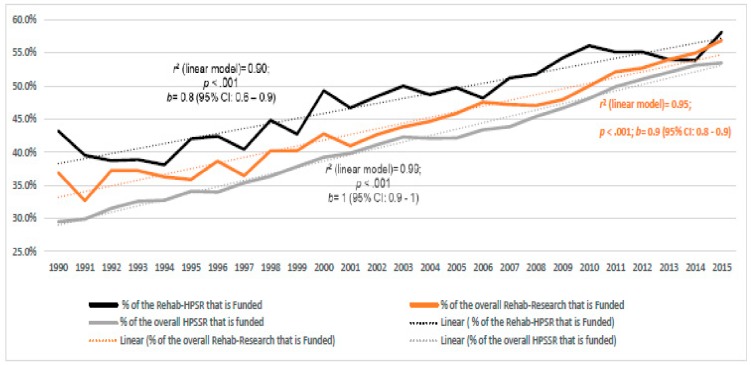
The yearly percentage of the research publications that were funded for the rehab (i.e., rehabilitation) Health Policy, Systems and Service Research (HPSSR), the overall rehab research, and the overall HPSSR (1990–2015), and the regression models that had the best fit with the data. The *p* values and 95% Confidence Intervals (CIs) refer to the linear regression model.

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
