# Peer review of "Development of the Rehabilitation Health Policy, Systems, and Services Research Field: Quantitative Analyses of Publications over Time (1990–2017) and across Country Type"

_ijerph, 2020, doi:10.3390/ijerph17030965_

Round 1

Reviewer 1 Report

This study provides a 30,000-foot snapshot of the relative growth in health policy, health systems, and health services research in the rehabilitation literature from 1990-2017. Given the dynamic healthcare reform environment over the past 15+ years, interest in and the impact of these three approaches have increased substantially. Thus, examining how the rehabilitation field has responded and evolved over this timeframe is a relevant research objective. While the simplistic methodology provides a vast overview and easy-to-replicate steps, the limitations are not trivial and should be seriously considered when interpreting the specific findings and potential implications. The following comments and suggestions are provided to improve clarity and better frame the limitations for IJERPH readers.

“Comparator fields” is mentioned in the Abstract and a couple of times in the Methods related to search filters, but it is never explicitly stated what those fields are. I assume they are rehabilitation as a whole and HPSSR as a whole, but clearly listing those as the comparator fields and using that language in the Results section would improve reader comprehension. Line 34: I have no idea what the second half of the Abstract Conclusion sentence is stating. Lines 46 and 54: Labeling health policy and health systems research as inter-disciplinary fields and health services research as a multidisciplinary field suggests tangible differences in the dynamics of these disciplines, yet the three are indistinct in the aggregated HPSSR category. Line 82: The perceived impact statement is overstated. The specific research questions have nothing to do with national and global research policies or with directly informing those policies. Rather, answering these questions will provide information on crude trends in HPSSR articles in the rehabilitation literature over the past 28 years. Lines 117-118: It may just be a typo, but this sentence is confusing. Line 150: I am no MeSH expert, but relying on the singular term “Support of Research” [Document Type] to classify studies as funded or not seems uncertain at best. I have no idea if all funding declarations included in the Acknowledgement section of some journals and Disclosure statements of other journals, etc., are automatically tagged with this MeSH term, but I am not confident. I suggest either confirming the thoroughness of the MeSH process for this term or doing a brief systematic check across different journals to confirm the accuracy of this dichotomous tag. Figures 1 and 3: If reporting the p value, parameter estimate, and 95% CI for the linear model in the figure, then that line should also be shown in that figure. Lines 237-238: Missing some values and/or words in this sentence. Line 245 (and Conclusions): It is difficult to understand how more than half of the publications were excluded through the country MeSH terms and how those excluded publications could dramatically impact these percentage calculations. Thus, one's confidence in this finding and interpretation is low. Yet, this is dominant theme in the Conclusions section. I suggest exploring the excluded articles (with the exception of the systematic reviews) to determine if they could in fact be classified based on the origin of the data with additional screening steps or if not, to provide more clarity for readers on the types of articles that were included in other calculations presented in this study but not in the country income calculations. Lines 307-308: Interesting interpretation with practical implications! Lines 320-321: These are two successive limitations, not off-setting one hand / other hand scenarios. Lines 344-348: I suggest reiterating that these findings are based on the total results of a crude search without the traditional subsequent screening steps to ensure content and fit. Further, given the uncertainty of the country focus MeSH noted previously, the LMIC finding should not dominate the Conclusions. Appendix 1 provides great detail and transparency. Grammar and typos need to be fixed throughout.

Author Response

Our reply to the reviewer 1 was uploaded as an attached document.

Reviewer 2 Report

Revision of manuscript:

DEVELOPMENT OF THE REHABILITATION HEALTH POLICY, SYSTEMS, AND SERVICES RESEARCH FIELD: QUANTITATIVE ANALYSES OF PUBLICATIONS OVER TIME (1990–2017) AND ACROSS COUNTRY TYPE

- Line 89, check word size in the sentence as well as the rest of article.

- In figure 1, the authors have the number of all rehabilitation research publications that was found in order to have the percentage of HPSSR publications? It seen the graph was built in y axis considering HPSSR percentage of rehabilitation research. For this reason it is important to present the number of this later publications in comparison to the HPSSR ones.

- This reviewer would like to address the following question to the authors: the analysis was made, as authors stated, in the comparison of rehabilitation researches (focused aspects) and HPSSR researches. The percentage and results shown was based on that logical comparative analysis. If HPSSR be analyzed in terms of its quantitative data found without rehabilitation research comparison, the results will be different from the ones obtained. Would this new results be indicative of other not spurious factors influencing HPSSR researches growth rather than trying to find the variance significance between these two considered parameters of the research (rehabilitation and HPSSR)?

- Line 321, grammar (, , )

- In discussion section, about the sentence “the rehabilitation of cognitive, communicative, and neuro-behavioral impairments as a result of or associated to physical impairments are included in the scope of rehabilitation covered”, do the authors have any glance of why HPSSR are growing in this field towards non rehabilitation journals? Would information systems related journals be absorbing these topics into a more computational and engineered medical related fields? Would this analysis in the discussion section be of importance for this research?

This reviewer feels the lack of correlation and references in discussion section about the empirical researches within the fields of physical impairments, cognitive, communicative, and neuro-behavioral impairments as a result of or associated to physical impairments with technological advancements of information system sciences in a broader way. This can be a valuable topic for rehabilitation researches publication discussions.

Author Response

We provide our reply to the reviewer 1's comments as an attached document.

Round 2

Reviewer 2 Report

Thank you for the review made by authors.

Besides the limited focus of research given and statistical tools to ensure no spurious correlations, the overall analysis can give a glance of types of publications and journals concerning rehabilitation researches.